# All-You-Can-Fit 8-Bit Flexible Floating-Point Format for Accurate and Memory-Efficient Inference of Deep Neural Networks

## Abstract

Modern deep neural network (DNN) models generally require a huge amount of weight and activation values to achieve good inference outcomes. Those data inevitably demand a massive off-chip memory capacity/bandwidth, and the situation gets even worse if they are represented in high-precision floating-point formats. Effort has been made for representing those data in different 8-bit floating-point formats, nevertheless, a notable accuracy loss is still unavoidable. In this paper we introduce an extremely flexible 8-bit floating-point (FFP8) format whose defining factors – the bit width of exponent/fraction field, the exponent bias, and even the presence of the sign bit – are all configurable. We also present a methodology to properly determine those factors so that the accuracy of model inference can be maximized. The foundation of this methodology is based on a key observation – both the maximum magnitude and the value distribution are quite dissimilar between weights and activations in most DNN models. Experimental results demonstrate that the proposed FFP8 format achieves an extremely low accuracy loss of $0.1\% \sim 0.3\%$ for several representative image classification models even without the need of model retraining. Besides, it is easy to turn a classical floating-point processing unit into an FFP8-compliant one, and the extra hardware cost is minor.

## 1 Introduction

With the rapid progress of deep neural network (DNN) techniques, innovative applications of deep learning in various domains, such as computer vision and natural language processing (NLP), are getting more mature and powerful (Huang et al., 2017; Vaswani et al., 2017; Szegedy et al., 2015; Howard et al., 2017; He et al., 2017; Krizhevsky et al., 2012).

To improve the model accuracy, one of the most commonly used strategies is to add more layers into a network, which inevitably increases the number of weight parameters and activation values of a model. Today, it is typical to store weights and activations in the 32-bit IEEE single-precision floating-point format (FP32). Those 32-bit data accesses thus become an extremely heavy burden on the memory subsystem in a typical edge or AIoT device, which often has very limited memory capacity and bandwidth. Even for high-end GPU or dedicated network processing unit (NPU) based computing platforms, off-chip DRAM bandwidth is still a major performance bottleneck.

To relieve the issue of memory bandwidth bottleneck, several attempts of various aspects have been made including (but not limited to) weight pruning (Li et al., 2017; Han et al., 2016), weight/activation quantization (Courbariaux et al., 2015; Hubara et al., 2017), and probably the most straightforward way: storing weights and activations in a shorter format (Köster et al., 2017). One trivial way to do so is to adopt the 16-bit IEEE half-precision floating-point format (FP16). An FP16 number consists of 1 sign bit, 5 exponent bits, and 10 fraction bits. In addition, Google proposed another 16-bit format, named Brain Floating-Point Format (BFP16), simply by truncating the lower half of the FP32 format (Kalamkar et al., 2019). Compared with FP16, BFP16 allows a significantly wider dynamic value range at the cost of 3-bit precision loss. Note that the exponent bias in all of the above formats is not a free design parameter. Conventionally, the value is solely determined by the exponent size. For example, for FP16 with 5-bit exponent, the exponent bias is automatically fixed to 15 ($2^{5-1} - 1$).

To make the data even shorter, 8-bit fixed-point signed/unsigned integer formats (INT8 and UINT8) are also broadly adopted. However, the 8-bit fixed-point format inherently has a narrower dynamic value range so that the model accuracy loss is usually not negligible even after extra symmetric or asymmetric quantization. As a consequence, there are a number of attempts concentrating on utilizing mixed-precision or pure 8-bit floating-point numbers in deep learning applications.

Various techniques have been developed for mixed-precision training (Banner et al., 2018; Micikevicius et al., 2018; Das et al., 2018; Zhou et al., 2016). Moreover, recent studies proposed several training frameworks that produces weights only in 8-bit floating-point formats (Wang & Choi, 2018; Cambier et al., 2020; Sun et al., 2019). In these studies, the underlying 8-bit floating-point numbers in training and inference are represented in the format of FP8(1, 5, 2) or FP8(1, 4, 3), where the enclosed three parameters indicate the bit length of sign, exponent, and fraction, respectively. Note that 4 or 5 bits are essential for the exponent in their frameworks, or the corresponding dynamic range may not cover both weight and activation values well. Consequently, only 2 or 3 bits are available for fraction, which inevitably leads to lower accuracy.

In this paper, we present an extremely flexible 8-bit floating-point (FFP8) number format. In FFP8, all parameters – the bit width of exponent/fraction, the exponent bias, and the presence of the sign bit – are configurable. Three major features of our inference methodology associated with the proposed FFP8 format are listed as follows. First, it is observed that both the maximum magnitude and the value distribution are quite dissimilar between weights and activations in most DNNs. It suggests the best exponent size and exponent bias for weights should be different from those for activations to achieve higher accuracy. Second, a large set of commonly-used activation functions always produce nonnegative outputs (e.g., ReLU). It implies that activations are actually unsigned if one of those activation functions is in use. Hence, it implies the sign bit is not required for those activations, which makes either exponent or fraction 1-bit longer. Note that even one bit can make a big impact since only 8 bits are available. Third, all aforementioned studies require their own sophisticated training frameworks to produce 8-bit floating-point models. Our flow does not. Our flow simply takes a model generated by any conventional FP32 training framework as the input. Then, it simply converts the given pre-trained FP32 model into an FFP8 model.

The rest of this paper is organized as follows. Section 2 briefly introduces related work. In Section 3, we elaborate more on the proposed FFP8 format and how to properly convert a pre-trained FP32 model into an FFP8 one. Section 4 demonstrates the experimental results on various DNN models. The system and hardware design issues for the support of FFP8 numbers are discussed in Section 5. Finally, the concluding remarks are given in Section 6.

## 2 RELATED WORK

DNNs are becoming larger and more complicated, which means they require a bigger memory space and consume more energy during inference. As a result, it is getting harder to deploy them on systems with limited memory capacity and power budget (e.g., edge devices). (Han et al., 2016; Zhao et al., 2020; Horowitz, 2014) also demonstrated that off-chip DRAM access is responsible for a significantly big share of system power consumption. Hence, it remains an active research topic about how to reduce the memory usage for weights and activations. As mentioned in the previous section, one way to do so is to use short 8-bit floating-point number formats.

Wang & Choi (2018) introduced a DNN training methodology using 8-bit floating-point numbers in FP8(1, 5, 2) format. The methodology features chunk-based accumulation and stochastic rounding methods for accuracy loss minimization. Besides, to achieve a better trade-off between precision and dynamic range during model training, Sun et al. (2019) proposed an improved methodology that utilizes two different 8-bit floating-point formats – FP8(1, 4, 3) for forward propagation and FP8(1, 5, 2) for backward propagation. Nevertheless, both methodologies fail to make a DNN model entirely in 8-bit numbers: the first and the last layers of the given model are still in 16-bit floating-point numbers; otherwise, the model suffers about 2% accuracy degradation. Cambier et al. (2020) then proposed the S2FP8 format, which allows a DNN model represented in 8-bit floating point numbers completely. By adding a scaling factor and a shifting factor, data can thus be well represented in FP8(1, 5, 2) after proper shifting and squeezing operations, which eliminates the need of 16-bit floating point numbers. However, the S2FP8 format still results in about 1% accuracy drop in ResNet-50 (He et al., 2018).

Figure 1: Various floating-point formats: (a) conventional ones, and (b) the proposed FFP8 format

# 3 FLEXIBLE 8-BIT FLOATING-POINT (FFP8) FORMAT

## 3.1 DEFINITION OF THE FFP8 FORMAT

In this subsection, a flexible 8-bit floating-point (FFP8) format, which leads to more accurate inference outcomes of deep neural networks, is presented. A typical floating-point format consists of sign ($s$), exponent ($e$), and fraction ($f$) fields, and the bit length of each field is specified as $x, y, z$, respectively. Besides, one more parameter, exponent bias ($b$), is required to completely specify a floating-point number. Conventionally, $b$ is always implicitly set to $2^{y-1} - 1$. In this paper, an $n$-bit floating-point format is denoted as $(x, y, z, b)$ or $(x, y, z)$, where $n = x + y + z$. If $b$ is missing, the default value is implicitly used.

Figure 1(a) illustrates one 16-bit FP16 format and two commonly used 8-bit formats: (1, 5, 2, 15) and (1, 4, 3, 7). Note that there is actually only one parameter, the bit width of the exponent ($y$), that can be freely chosen when defining a new $n$-bit conventional floating-point format since $x$ is always 1, $z$ is always $n - y - 1$, and $b$ is always $2^{y-1} - 1$. The above fact motivated us to think out-of-the-box and thus develop a more flexible format, named flexible 8-bit floating-point (FFP8) format and shown in Figure 1(b). In addition to the size of exponent ($y$), the FFP8 format offers two more parameters. First, the exponent bias ($b$) is not necessarily equal to $2^{y-1} - 1$ and can be set to any integer, which helps cover the value distributions of both weights and activations well in a shorter exponent size ($y$). Second, the sign bit is present or not (i.e., $x$ can be 0 or 1). Without the sign bit ($x = 0$), unsigned activations can be better represented in higher precision. To be more precise, there are only two restrictions on an $n$-bit FFP8 format: 1) $x + y + z = n$; 2) $x$ must be 0 or 1. Next, we are about to show how the proposed FFP8 format improves the inference accuracy.

## 3.2 WEIGHT DISTRIBUTION AND THE WAYS VARIOUS 8-BIT FORMATS COPE WITH IT

Overall speaking, the magnitudes of weights in most DNN models are usually small. Take a popular image classification model VGG-16 (Simonyan & Zisserman, 2015) as an example, the maximum magnitude of weights in the whole model is less than 2. Figure 2 gives the overall weight distribution (in log scale) of VGG-16 trained via a conventional FP32 framework. Figure 2(a) illustrates how the conventional FP8(1, 4, 3) copes with those weights. The rectangle in red is called the range window, which specifies the value range that FP8(1, 4, 3) can represent. The purple vertical dash line further partitions the window into the norm region (right side) and the denorm region (left side). The star marks the position where the weight of the maximum magnitude locates. Note that virtually the entire right half of the range window in Figure 2(a) covers no weights, while 9.6% of leftmost weights cannot be included in the range window. In order to contain almost every weight in a range window, FP8(1, 5, 2) can be selected alternatively, as shown in Figure 2(b). A bigger exponent size (4 to 5) results in a larger range window. However, there are only 4 ($2^2$) instead of 8 ($2^3$) representative values available for each $x$ in the norm region since the fraction size decreases from 3 to 2, which potentially results in a lower accuracy. Note that FP8(1, 4, 3) and FP8(1, 5, 2) are two most commonly used 8-bit floating point formats in the previous studies.

With the help of the proposed FFP8 format, things can change a lot. Figure 2(c) shows what happens if FFP8(1, 4, 3, 15) is in use. The range window is of the same size as that in Figure 2(a) but is left-shifted by 8 positions due to the exponent bias is set to 15 instead of the default value 7. It is obvious that FFP8(1, 4, 3, 15) can better cope with weights in VGG-16 than FP8(1, 4, 3) and FP(1, 5, 2). Furthermore, Figure 2(d) shows what if FFP8(1, 3, 4, 7) is in use. Comparing FFP8(1, 3, 4, 7) against FFP8(1, 4, 3, 15), weights in the norm region (over 35%) are represented in 1-bit

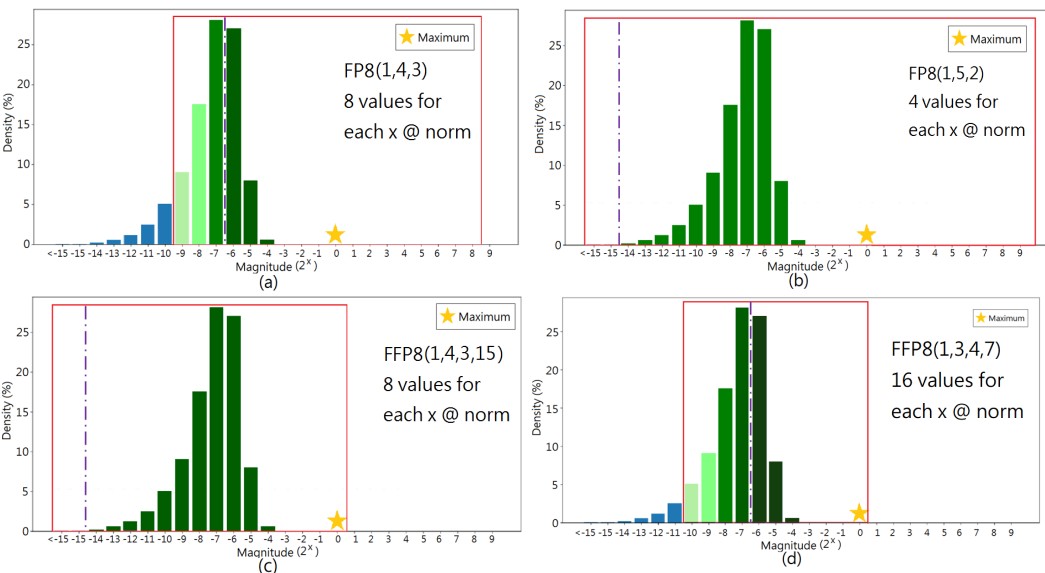

Figure 2: The range windows of various 8-bit floating-point formats versus the weight distribution of VGG-16; (a) in FP8(1, 4, 3), (b) in FP8(1, 5, 2), (c) in FFP8(1, 4, 3, 15), (d) in FFP8(1, 3, 4, 7)

higher precision whereas 4.6% of leftmost weights are not included in the range window. However, those out-of-the-window weights can be regarded as some sort of pruned weights. That is, more investigation should be further conducted to determine whether FFP8(1, 3, 4, 7) or FFP8(1, 4, 3, 15) is better for the weights in VGG-16. One way or another, it is clear that the proposed FFP8 formats can achieve certain improvement that the conventional 8-bit formats cannot.

## 3.3 FFP8-BASED INFERENCE FRAMEWORK AND BEST-FIT FORMAT SELECTION

In the proposed memory-efficient inference framework, weights and activations stored in off-chip memory are in FFP8 formats while operations inside the computing engine are still in FP32, which is a common strategy widely adopted in current state-of-the-art computing platforms (Intel, 2018; Nvidia, 2017). Hence, a systematic process, shown in Figure 3, is required to transform FP32 numbers into FFP8 ones in the following two scenarios: 1) quantizing pre-trained FP32 weights to FFP8 ones in advance, and 2) converting FP32 output activations into FFP8 ones before writing them back to off-chip memory. The process first prepares a list of 256 values that can be precisely represented by the specified FFP8 format. Then, the given FP32 inputs (weights or activations) are converted into FFP8 outputs using the round-to-nearest-even method.

In order to explore best-fit 8-bit formats for weights and activations in VGG-16, we have made a set of attempts, and the results are summarized in Table 1. First, it is unwise to represent weights in FP(1, 5, 2). Though Figure 2(b) shows FP(1, 5, 2) can cover virtually all weights, the accuracy loss is serious due to its 2-bit extremely low precision. Next, FP(1, 4, 3) performs much better than FP(1, 5, 2), which suggests one extra precision bit does make a notable improvement here though 9.6% smallest weights are clear to 0 (i.e., pruned away), as depicted in Figure 2(a). Moreover, the proposed FFP8 format can surely do better. FFP8(1, 4, 3, 15) further outperforms FP(1, 4, 3) because weights are better covered by its left-shifted range window, as shown in Figure 2(c).

Inspired by the fact that FP(1, 4, 3) outperforms FP(1, 5, 2), we further examine whether an even smaller exponent size can help or not. First, it is unwise to represent weights in FP8(1, 3, 4) without modifying the default exponent bias (=3). Though the fraction size increases from 3 to 4, 64.1% leftmost weights are out of the range window, which is way too much. However, by setting the exponent bias to 7, which is equivalent to sliding the range window to the left by 4, the resultant FFP8(1, 3, 4, 7) successfully achieves a higher accuracy since it has a bigger 4-bit fraction size and only prunes away 4.6% smallest weights, as illustrated in Figure 2(d).

Figure 3: The process converting FP32 inputs into FFP8 outputs with the given format parameters

Table 1: Top-1 and Top-5 accuracy of VGG-16 on ImageNet dataset (Deng et al., 2009) in various number formats; delta ($\Delta$) indicates the accuracy drop as compared to FP32

| Weight | Activation | Top-1 ($\Delta$) | Top-5 ($\Delta$) |
|---|---|---|---|
| FP32 | FP32 | 71.59% | 90.38% |
| (1,5,2,15) | (1,4,3,7) | 69.89% (-1.70%) | 89.29% (-1.09%) |
| (1,4,3,7) | (1,4,3,7) | 70.86% (-0.73%) | 90.02% (-0.36%) |
| (1,4,3,15) | (1,4,3,7) | 70.96% (-0.63%) | 90.10% (-0.28%) |
| (1,3,4,3) | (1,4,3,7) | 70.18% (-1.41%) | 89.56% (-0.82%) |
| (1,3,4,7) | (1,4,3,7) | 71.19% (-0.40%) | 90.12% (-0.26%) |
| (1,3,4,7) | (1,4,3,7)+(0,4,4,7) | 71.19% (-0.40%) | 90.14% (-0.24%) |

In addition to weights, it is certainly worth finding out best-fit formats for activations as well. For those attempts made above, activations are always in FFP8(1, 4, 3, 7) for two reasons: 1) the overall distribution of activations is wider than that of weights, and 2) the maximum magnitude of activations is much bigger than that of weights. The detailed distribution of activations will be given in Section 3.4 later. Meanwhile, it is also worth noting that a large set of commonly used activation functions always produce nonnegative outputs, e.g., ReLU, ReLU6, and sigmoid. That is, if those outputs are represented in any signed format, a half of the code space is actually wasted. It may not be a problem for 32-bit and 16-bit formats with long enough fraction bits; however, it is indeed a serious issue for any 8-bit format, which merely has 256 available codes in total. Since VGG-16 utilizes ReLU as its activation function, it is feasible to select signed FFP8(1, 4, 3, 7) for the first layer and unsigned FFP8(0, 4, 4, 7) for all succeeding layers. It implies that 256 instead of 128 codes are available to represent those unsigned activations in all layers except for the first one. With no surprise, the accuracy is further improved since the range window remains untouched while the fraction size gains one extra bit. In the last configuration, the Top-1 accuracy loss is only 0.4% when compared against the FP32 baseline, as indicated in Table 1.

## 3.4 LAYER-WISE OPTIMIZATION

Figure 4 and Figure 5 illustrate several distributions of weights and activations in VGG-16, respectively. Each figure includes the distributions of one whole model and three selected individual layers. In Section 3.3, the best-fit FFP8 format is determined by the overall distribution of the whole model. Here we have three key observations from those distributions: 1) the distributions of weights are quite dissimilar to those of activations, 2) even the distributions across different layers are dissimilar for both weights and activations, and 3) the distribution of an individual layer is narrower than that of the whole model. The above observations clearly suggest that applying layer-wise optimization (LWO) properly on number format selection is very likely to improve the accuracy further.

For instance, after comparing Figure 5(a)&(b), it is found that the distribution of the whole model is wider than that of the first layer, and the maximum log magnitudes of the whole model and the first layer are 8 and 1, respectively. As a consequence, selecting FFP8(1, 3, 4, 6) instead of FFP8(1, 4, 3, 7) for activations in the first layer can further increase the Top-1 accuracy by 0.19% (from 71.19% to 71.38%), as indicated in Table 2. Next, we examine a new configuration that makes weights of all layers are in FFP8(1, 2, 5, 3). It is obvious an unwise attempt since 37.6% leftmost weights are out of the range window according to Figure 4(a). However, it may not be a bad idea to use FFP8(1, 2, 5, 5) in Layer 6 and FFP8(1, 2, 5, 6) in the last layer because only 6.9% and 5.3% smallest weights are out of the range window respectively according to Figure 4(c)&(d). Therefore, we examine another new configuration that makes weights of all layers in FFP8(1, 2, 5, *). Here the asterisk (*) represents the largest possible exponent bias, which ensures that the maximum weight is still inside

the range window. This LWO on weights successfully increases the Top-1 accuracy by 0.1% (from 71.38% to 71.48%). Similarly, we can apply the LWO on activations, which again raises the Top-5 accuracy by 0.05%. The selected FFP8 format of each layer after LWO is reported in Table 3.

Previous studies usually choose FP8(1, 4, 3) or FP8(1, 5, 2) formats because the exponent size has to be large enough to cover most weights and activations at the cost of an even smaller fraction size. However, the exponent size for weights can be as small as 2 after LWO in our flow. Consequently, the larger 5-bit fraction size does help in accuracy improvement, as indicated in Table 2.

Note that the Top-1 accuracy achieved by the final configuration, which applies layer-wise optimization on both weights and activations, is merely 0.11% lower as compared to that of the FP32 baseline (71.48% vs. 71.59%). More importantly, model retraining is not applied yet. In other words, all the accuracy improvements made so far are simply from properly re-expressing weights and activations of a pre-trained FP32 model in their best-fit FFP8 formats.

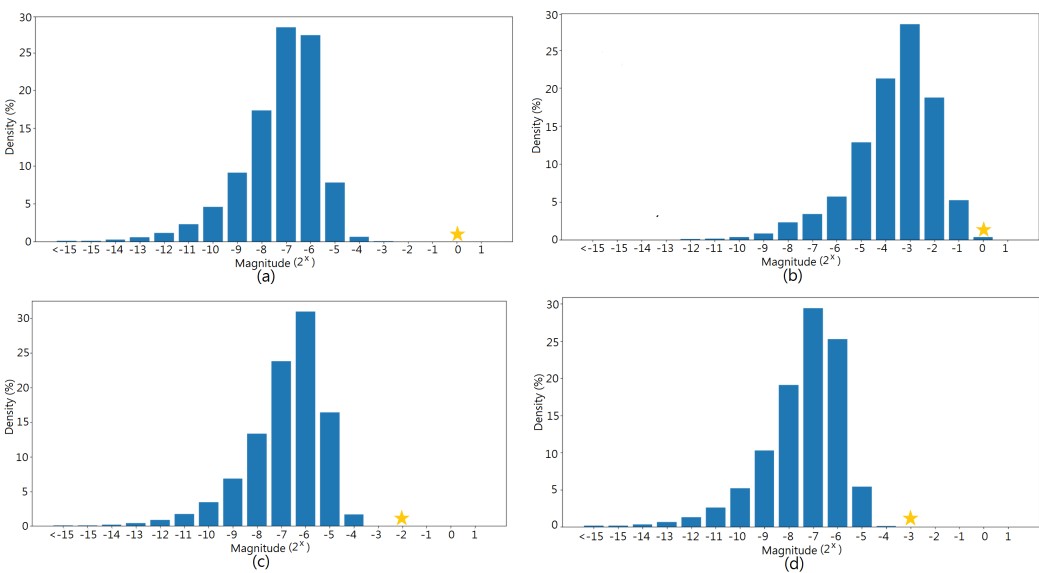

Figure 4: Weight distributions of (a) whole model, (b) first layer, (c) Layer 6, (d) last layer

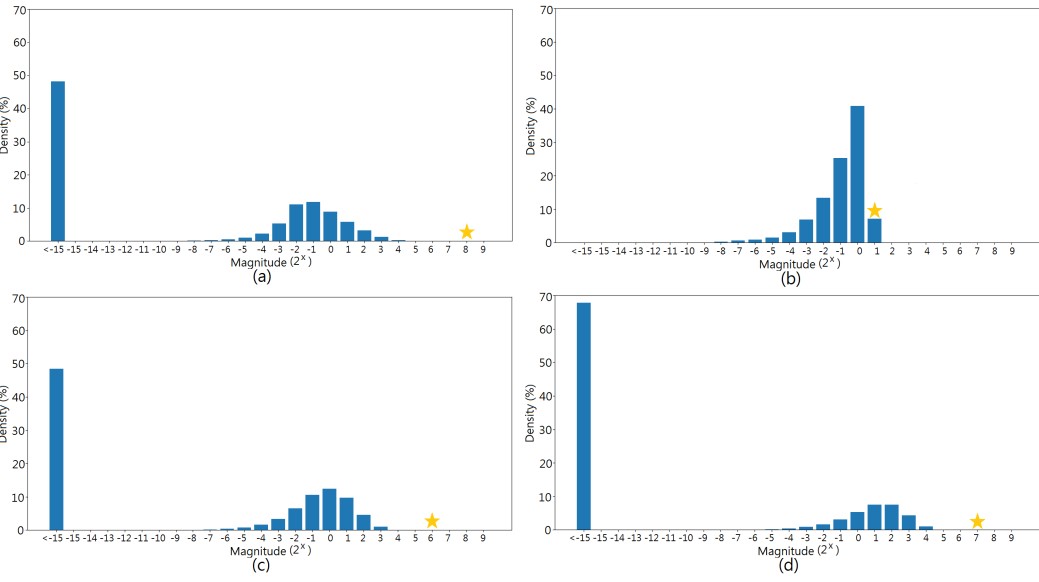

Figure 5: Activation distributions of (a) whole model, (b) first layer, (c) Layer 6, (d) last layer

Table 2: Top-1 and Top-5 accuracy of VGG-16 after layer-wise optimization

| Weight | Activation | Top-1 ($\Delta$) | Top-5 ($\Delta$) |
|---|---|---|---|
| FP32 | FP32 | 71.59% | 90.38% |
| (1,3,4,7) | (1,4,3,7)+(0,4,4,7) | 71.19% (-0.40%) | 90.14% (-0.24%) |
| (1,3,4,7) | (1,3,4,6)+(0,4,4,7) | 71.38% (-0.21%) | 90.33% (-0.05%) |
| (1,2,5,3) | (1,3,4,6)+(0,4,4,7) | 71.24% (-0.35%) | 90.22% (-0.16%) |
| (1,2,5,*) | (1,3,4,6)+(0,4,4,7) | 71.48% (-0.11%) | 90.27% (-0.11%) |
| (1,2,5,*) | (1,3,4,6)+(0,4,4,*) | 71.48% (-0.11%) | 90.32% (-0.06%) |

Table 3: The FFP8 format of each layer in VGG-16 after layer-wise optimization

| Layer | Weight | Activation | Layer | Weight | Activation |
|---|---|---|---|---|---|
| 1 | (1,2,5,3) | (1,3,4,6) | 8 | (1,2,5,5) | (0,4,4,8) |
| 2 | (1,2,5,4) | (0,4,4,11) | 9 | (1,2,5,5) | (0,4,4,8) |
| 3 | (1,2,5,4) | (0,4,4,10) | 10 | (1,2,5,6) | (0,4,4,8) |
| 4 | (1,2,5,5) | (0,4,4,10) | 11 | (1,2,5,5) | (0,4,4,7) |
| 5 | (1,2,5,4) | (0,4,4,9) | 12 | (1,2,5,5) | (0,4,4,7) |
| 6 | (1,2,5,5) | (0,4,4,9) | 13 | (1,2,5,6) | (0,4,4,8) |
| 7 | (1,2,5,4) | (0,4,4,9) | | | |

## 3.5 ACCURACY IMPROVEMENT VIA MODEL RETRAINING

All FFP8 weights in our previous experiments are simply converted from pre-trained weights generated from a typical FP32 training framework. However, it is reported that quantization-aware model retraining can usually improve the accuracy (Jacob et al., 2017), which motivates us to check whether it can successfully apply to our work. We first train the ResNet-18 model in an FP32 framework and the corresponding Top-1 accuracy is 69.76%. Next, an FFP8-based inference is performed under the following settings: 1) all weights are in FFP8(1, 3, 4, 7), 2) activations of the first layer are in FFP8(1, 3, 4, 6), and 3) activations of the other layers are in FFP8(0, 4, 4, 7). The Top-1 accuracy for the above configuration is down to 69.44%. Then, a quantization-aware retraining process of 15 epochs is applied and the resultant Top-1 accuracy goes back to 69.76%. Note that the retraining is done simply in a typical FP32 framework without the need of special training skills.

## 4 EXPERIMENTAL RESULTS ON VARIOUS DNN MODELS

In this section, we intend to demonstrate that the proposed FFP8 format constantly performs well in various DNN models in addition to VGG-16. Table 4 reports the accuracy results of various models under different format configurations. Configuration A gives the results of the FP32 baseline. Configuration B utilizes a conventional FP8(1, 4, 3) format with a default exponent bias (7), which incurs a penalty of roughly 1% drop on Top-1 accuracy. With the help of the FFP8 format, Configuration C adopts the best-fit format for weights, which effectively reduces the Top-1 accuracy loss to 0.4~0.75%. Configuration D further selects two best-fit formats for activations (the signed one for the first layer and the unsigned one for the rest), which minimizes the Top-1 accuracy loss to 0.3%. Note that neither layer-wise optimization nor model retraining is even applied for this achievement.

Table 4: Accuracy results of various models under different format configurations

| Cfg. | Weight | Activation | VGG-16 Top-1 / Top-5 | ResNet-50 Top-1 / Top-5 | ResNet-34 Top-1 / Top-5 | ResNet-18 Top-1 / Top-5 |
|---|---|---|---|---|---|---|
| A | FP32 | FP32 | 71.59 / 90.38 | 76.13 / 92.86 | 73.31 / 91.42 | 69.76 / 89.08 |
| B | (1,4,3,7) | (1,4,3,7) | 70.86 / 90.02 | 75.24 / 92.52 | 72.39 / 90.97 | 68.70 / 88.50 |
| C | (1,3,4,7) | (1,4,3,7) | 71.19 / 90.12 | 75.38 / 92.68 | 72.81 / 91.16 | 69.25 / 88.80 |
| D | (1,3,4,7) | (1,3,4,6)+(0,4,4,7) | 71.38 / 90.33 | 75.85 / 92.81 | 73.12 / 91.33 | 69.44 / 88.93 |

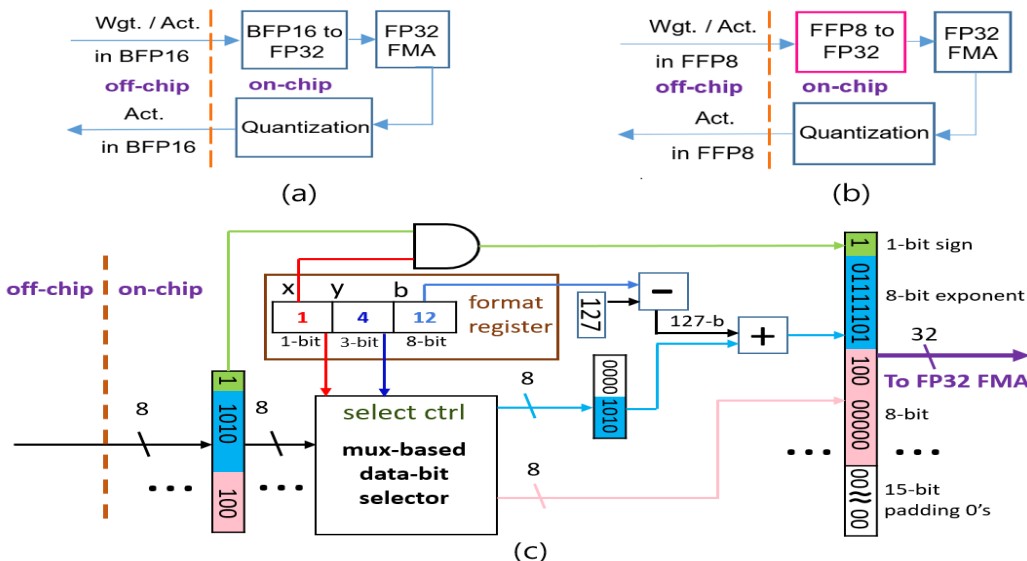

Figure 6: Memory-efficient system architectures: off-chip external data (a) in BFP16, (b) in FFP8; and (c) an FFP8 to FP32 hardware converter

In addition to image classification, we also want to know whether the proposed FFP8 format performs equally well in other application domains. Here, we examine two more applications: semantic segmentation and ECG check. First, FCN32s is a popular CNN model for semantic segmentation (Long et al., 2014). If FCN32s is in FP32, the mIOU is 63.63% using the VOC2011 dataset (Everingham et al.). Alternatively, if all the weights are in FFP8(1, 4, 3, 14) and all the activations are in FFP8(1, 4, 3, 2), the resultant mIOU would be 63.45%, a slight drop of 0.18%. Second, an LSTM model for ECG check (Physionet, 2017), has also been tested. If the LSTM model is in FP32, the check accuracy is 81.12%. If all the weights are in FFP8(1, 3, 4, 6), the first-layer activations are in FFP8(1, 3, 4, 5), and activations in the other layer are in FFP8(1, 4, 3, 16), the resultant accuracy would be 81.53%, an accuracy gain of 0.41%. The experimental results once again demonstrate that FFP8 performs very well in these two categories as well.

## 5 ASPECTS OF SYSTEM AND HARDWARE

The target of this work is to develop a memory-efficient inference system, especially for those with limited memory capacity and bandwidth. A current state-of-the-art system, proposed by Nvidia and Intel, has successfully cut the required memory size and traffic through representing external data (weights or activations) in BFP16/INT8 instead of FP32, as illustrated in Figure 6(a) (Intel, 2018; 2019; Nvidia, 2017; 2020). It not only reduces the memory size, alleviates the performance bottleneck due to memory bandwidth limitation but also saves a significant amount of energy due to fewer power-consuming external memory access operations. To preserve the computation accuracy at the same time, external BFP16 data are converted to FP32 data right before entering the FP32 fused-multiply-add (FMA) unit. That is, internal computations can all be in FP32. Those FP32 data are converted back to BFP16 only if they are about to be written back to external memory.

In this work, we propose a system architecture that is very similar to the previous one, as shown in Figure 6(b). The key difference is that external data are in FFP8 instead of BFP16, which implies the proposed system merely demands a quarter of the memory size and throughput required by today's FP32-based counterparts.

It is easy to convert a BFP16 number into its FP32 equivalent by concatenating a 16-bit pattern of all 0s at its least significant end. In fact, it is also easy to transform an FFP8 number to its FP32 equivalent via a converter, as depicted in Figure 6(c). Few register bits are allocated to store the current format settings of $x$, $y$, and $b$ within the converter. Then, $x$ is used to recover the sign bit; the biased exponent and fraction can be extracted via $x$ and $y$; finally the exponent can be further

corrected by the exponent bias $b$. Hence, it is apparent that the required hardware logic for the converter is indeed minimal and the extra hardware cost is truly minor as well. Furthermore, updates of those format registers are extremely infrequent. Even the layer-wise optimization is applied, those registers are only modified at the start of each layer. In other words, virtually no runtime overhead is imposed due to those register updates.

## 6 CONCLUSION

In this work, we propose the flexible 8-bit floating-point (FFP8) format for accurate and memory-efficient inference of deep neural networks. Our FFP8 format offers three adjustable options: 1) the size of exponent/fraction field, 2) the value of exponent bias, and 3) the presence of the sign bit, whereas those rigid conventional formats simply leave nothing. In this paper, we explain how the exponent size and bias jointly define the representable value range of a given FFP8 format. We also demonstrate how to explore the best-fit signed/unsigned FFP8 formats for weights and activations to achieve more accurate inference outcomes. Besides, a layer-wise optimization flow, which discovers the best-fit formats for each individual layer, is presented to further improve the accuracy. A model retraining methodology that can be carried out in typical FP32 frameworks without the need of special training skills is also introduced. The experimental results on various DNN models show that the proposed FFP8-based inference flow achieves an extremely low accuracy loss of $0.1\% \sim 0.3\%$ as compared to the FP32 baseline even without model retraining. Moreover, we also show that the extra hardware for supporting the FFP8 format is minimal. Therefore, it is conclusive that the proposed FFP8-based inference framework should be a better solution for those computing systems with limited memory capacity and bandwidth, e.g., edge and AIoT devices.

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
