# OpenReview forum: "All-You-Can-Fit 8-Bit Flexible Floating-Point Format for Accurate and Memory-Efficient Inference of Deep Neural Networks"
_ICLR.cc/2021/Conference — Reject_

### Official Review · AnonReviewer2 · 2020-10-28
**More information about HW and Compiler support needed**

**Rating:** 4
**Confidence:** 4

**Review:**

## Summary:

The paper introduces a new floating point format FFP8 that can adaptively choose the exponent bias as well as the existence of the sign bit. FFP8 is more flexible than other 8-bit floating point formats with fixed exponent biases. The authors show how FF8 can be used to cover the dynamic range of NN weights and activations while still achieving higher precision than commonly used FP 8 formats.

## Strengths:

* Experiments on how various settings of FFP8 can be used to represent weights and activations of different layers and comparison with existing FP 8 formats.
* Varying exponent bias is a good insight to shift the dynamic range of FP formats.

## Weaknesses and Questions for the Authors

* Limited details on how hardware support for FFP8 look like.
The authors claim in section 5 that the extra hardware support required for FFP8 would be minimal. However, no data or simulations are there to back this claim. In a possible hardware implementation of FFP8, each floating point instruction should be preceded by an instruction which sets (x,y,b) registers for FFP8 to F32 conversion. The frequency of setting these registers may vary based on the application, but nevertheless is an overhead compared to the current FP 8 implementations. The authors should quantify this overhead with a hardware simulation. Alternatively, the authors can describe a different and an efficient implementation of FP 8. This is an important detail that is missing from the paper.
* Compiler support for FFP8
 The authors have not described how FFP 8 would be emitted, specially when different settings are used for different layers. Do you expect the programmer to hand code this? Or will there be compiler support? If so, what would the compiler support look like? What would the compiler optimization algorithm look like to produce FFP 8 code? These details are not answered by the paper.
* More evaluation on other commonly used NNs
The authors present results for VGG-16. I would like to see more results on different networks. The authors can cut down on some of the graphs or make them smaller to get more space (specially Figure 5). Also, in general I feel like rather than giving out all the results that bring us to the same conclusion, the authors can summarize the results or graphs to make the message / conclusion of the paper more clearer.

---

> ### Author Response · Authors · 2020-11-16
> **Reply to AnonReviewer 2 Part 3**
>
> << Summary Section>>
>
> In the comments, the reviewer has raised four major concerns and requested for more information and explanation:
>
> (1) The hardware support details for FFP8  =>  Q1
>
> (2) The frequency of setting format registers  =>  Q2
>
> (3) The compiler/framework support for FFP8 and how the programmers can adopt FFP8 easily in a framework  =>  Q3
>
> (4) More evaluation results on other NNs  =>  Q4
>
> In the above response, we have addressed all the reviewer’s concerns in detail and provided the corresponding answers each by each. We sincerely hope the reviewer can re-evaluate the contribution of this paper after examining our answers to all the raised questions here. Thank you very much.

---

> ### Author Response · Authors · 2020-11-16
> **Reply to AnonReviewer 2 Part 2**
>
>
> Q3:
> Compiler support for FFP8. The authors have not described how FFP 8 would be emitted, especially when different settings are used for different layers. Do you expect the programmer to hand code this? Or will there be compiler support? If so, what would the compiler support look like? What would the compiler optimization algorithm look like to produce FFP 8 code? These details are not answered by the paper.
>
> A:
> Thank you very much for bringing up the issue of compiler support. We elaborate more on this topic as follows.
>
> We certainly do not expect the programmers to deal with the details about how to correctly convert an FFP8 number to/from a typical FP32 number. If we do, the proposed method would be programmer-unfriendly. Instead, in order to be programmer-friendly, those details must be hidden behind the supports of compilers and deep learning frameworks (e.g., TensorFlow, PyTorch, ...).
>
> There may be a lot of feasible approaches to accomplish this goal and we just show one of the possible ways. The following shows a hypothetical code segment, which demonstrates what would look like if the proposed FFP8 format is supported in PyTorch:
>
> **********
> \# CNN Layer 1 of VGG-16
>
> self.conv1_1 = nn.Conv2d(3, 64, 3, padding=1)
>
> conv1_1.ffp8(w_config = (1,2,3), in_config = (1,3,6), out_config = (0,4,11))
>
> self.relu1_1 = nn.ReLU(inplace=True)
>
> \# CNN Layer 2 of VGG-16
>
> self.conv1_2 = nn.Conv2d(64, 64, 3, padding=1)
>
> conv1_2.ffp8(w_config = (1,2,4), in_config = (0,4,11), out_config = (0,4,10))
>
> self.relu1_2 = nn.ReLU(inplace=True)
>
> self.pool1 = nn.MaxPool2d(2, stride=2, ceil_mode=False)
>
> ...
>
> \# CNN Layers 3~12 of VGG-16 here
>
> ...
>
> \# CNN Layer 13 of VGG-16
>
> self.conv5_3 = nn.Conv2d(512, 512, 3, padding=1)
>
> conv5_3.ffp8(w_config = (1,2,6), in_config = (0,4,8), out_config = (0,4,8))
>
> self.relu5_3 = nn.ReLU(inplace=True)
>
> self.pool5 = nn.MaxPool2d(2, stride=2, ceil_mode=False)
>
> ...
> **********
>
> In the above code segment, you can find three statements including a substring of “ffp8”. Each of them explicitly specifies the FFP8 format parameters (x, y, b) of weights, input activations, and output activations for the corresponding CNN layer, respectively. That is, each of such function call is designated to properly set the hardware “format registers” with the help of compilers and/or learning frameworks. Note that those function calls do nothing but set format registers only. If those ffp8-related function calls are removed, the above code segment looks just like a regular code segment for running VGG-16 in the current PyTorch environment.
>
> In this way, the compiler/framework developers can easily provide the FFP8-related supports, while the compiler/framework users can easily adopt the FFP8 data format in their deep learning applications with minimal extra effort.
>
> ====================
>
>
> Q4:
> More evaluation on other commonly used NNs. The authors present results for VGG-16. I would like to see more results on different networks. The authors can cut down on some of the graphs or make them smaller to get more space (especially Figure 5). Also, in general I feel like rather than giving out all the results that bring us to the same conclusion, the authors can summarize the results or graphs to make the message / conclusion of the paper more clearer.
>
> A:
> Thank you very much for the valuable comment.
> In our first submission, we reported the results for not only VGG-16 but also ResNet-18/34/50. Currently, in addition to the models mentioned above, we have more results about FCN32s for semantic segmentation, and an LSTM model for ECG check.
>
> FCN32s is a popular network model for semantic segmentation. For FCN32s in default FP32, the mIOU is 63.63% using the VOC2011 dataset. Alternatively, if all the weights are in FFP8(1,4,3,14) and all the activations are in FFP8(1,4,3,2), the resultant mIOU is 63.45%, a slight drop of 0.18%.
>
> Besides, an LSTM model for ECG examination, [ https://physionet.org/content/challenge-2017/1.0.0/ ], has also been evaluated. For the LSTM model in default FP32, the check accuracy is 81.12%. If all the weights are in FFP8(1,3,4,6), the first-layer activations are in FFP8(1,3,4,5), and activations in the other layer are in FFP8(1,4,3,16), the resultant accuracy is 81.53%, an ACCURACY GAIN of 0.41%.
>
> Currently, we keep on evaluating more neural network models in various application domains in additional to image classification. For example, one of the above two models is a CNN for semantic segmentation while the other is an LSTM for ECG examination. The experimental results have demonstrated that FFP8 also work very well in these two categories.
>
> We will include these new results into Section 4 of the revised manuscript to demonstrate that the proposed FFP8 format can be applied to a broad range of deep learning applications.

---

> ### Author Response · Authors · 2020-11-16
> **Reply to AnonReviewer 2 Part 1**
>
> Q1:
> Limited details on how hardware support for FFP8 look like. The authors claim in section 5 that the extra hardware support required for FFP8 would be minimal. However, no data or simulations are there to back this claim.
>
> A:
> Thank you for your valuable comment.
> Due to the page limitation, we only put a very simplified hardware block diagram in Figure 6(c) of our initial manuscript. According to your suggestion, more hardware details of the proposed converter is revealed as shown in [ https://imgur.com/a/CQam38l ], and the manuscript will be further revised to include this new figure accordingly.
>
> As illustrated above, the extra hardware merely consists of few register bits, few logic gates, a mux-based data-bit selector, a subtractor, and an adder. An example of conversion from FFP8 to FP32 is also included. It should be clear enough that the size of the proposed converter is indeed much smaller than that of a 32-bit floating-point fused-multiply-add (FMA) unit, and thus the extra hardware support is considered minimal.
>
> ====================
>
> Q2:
> In a possible hardware implementation of FFP8, each floating point instruction should be preceded by an instruction which sets (x,y,b) registers for FFP8 to F32 conversion. The frequency of setting these registers may vary based on the application, but nevertheless is an overhead compared to the current FP 8 implementations. The authors should quantify this overhead with a hardware simulation. Alternatively, the authors can describe a different and an efficient implementation of FP 8. This is an important detail that is missing from the paper.
>
> A:
> Thank you very much for the comment. Please allow us to elaborate more on this issue.
>
> We would like to point out that the statement “each floating point instruction should be preceded by an instruction which sets (x,y,b) registers for FFP8 to F32 conversion” is not correct. In fact, the “format registers” shown in [ https://imgur.com/a/CQam38l ] are updated only when a new FFP8 format is selected for the incoming activations or weights. That is, the format registers are set
> ONLY ONCE if a deep learning application only adopts one format for activation and one format for weight throughout the entire application time. If the proposed layer-wise optimization (LWO) technique is in use, those “format registers” are updated ONLY at the beginning of each layer. Take VGG-16 as an example, there are more than 1.8 billon MAC operations in Layer 2. It suggests once the format registers are properly set, those settings remain unchanged and valid for the succeeding computational operations for a long while. Moreover, there are at least 90 million MACs in any layer of VGG-16. In summary, the update of format registers is indeed infrequent and the runtime overhead for setting the format registers is negligible.
>
> Let us emphasize once again: the update of the format register is infrequent. Once set, all succeeding activation and weight values from the external memory are simply 8-bit FFP8 data transfers, as the figure demonstrates.

---

### Official Review · AnonReviewer1 · 2020-10-28
**paper lacks clear objective and novelty**

**Rating:** 3
**Confidence:** 5

**Review:**

This paper explores 8-bit floating point formats for the inference of deep neural networks. The quantization is applied on the weight and activation tensors, but computation engine remains in FP32.  To cover the different ranges of weight and activation tensors, the authors propose to use exponent bias. The authors did ablation studies on the impact of numerical formats, e.g. bit-width and exponent biases, on model accuracies. The experiments are performed on vision models on ImageNet, including VGG and ResNet 50/34/18.

The paper is relatively clear written, and experiments seem sound, however, I have some major concerns on the objective and novelty of this paper.

1). It is not clear to me the objective of this paper. This paper introduces an 8-bit quantized inference framework. However, it is well-known that, 8-bit precision can be applied to popular DNN models to accelerate inference while maintaining model accuracies and many such systems have already been put into products (e.g. TPU). The state-of-the-art inference quantization work are primarily in the precision of 4 bit or less. But this paper did not compare their work to any of the latest inference works. Instead, the three references this paper compared to, i.e. (Wang & Choi, 2018; Cambier et al., 2020; Sun et al., 2019), are all for the acceleration of DNN TRAINING, where the challenging can be very different, e.g. quantization of gradient tensors.

2). On the same note, this paper claimed advantages of using FP32 computation engine, however, the three training papers, by definition, can also be used in FP32 computations through the same high precision converting described in this paper. In addition, it is not clear to me how much one can benefit from quantization while keeping computation in 32bit? For example, one may save some memories space, but the whole inference will be limited by the FP32 computations in terms of throughput, latency, power and cost. Can the authors provide an application case where only quantizing data is beneficial?

3). The main contribution claimed in the paper are to use exponent bias to cover tensors with different range of distribution. However, this technique has be introduced and discussed in (Sun et at., 2019) paper and has also been investigated in detail in (Cambier et al., 2020) paper. The authors of this paper did not provide any new insight although using in a much simpler case, i.e. only forward paths.

I think this paper lacks clear objective, novelty and insightful analysis, therefore not good enough for ICLR.

---

> ### Author Response · Authors · 2020-11-17
> **Reply to AnonReviewer 1 Part 3**
>
> Q3:
> The main contribution claimed in the paper are to use exponent bias to cover tensors with different range of distribution. However, this technique has be introduced and discussed in (Sun et al., 2019) paper and has also been investigated in detail in (Cambier et al., 2020) paper. The authors of this paper did not provide any new insight although using in a much simpler case, i.e. only forward paths.
>
> A:
> Thank you for the comment. We respectfully disagree with the reviewer, however. Please allow us to elaborate more on this issue.
>
> 1)
> In (Sun et al., 2019), the authors use FP(1,4,3) for forward inference and FP(1,5,2) for backward propagation. For forward inference, a nontrivial exponent bias value 11 instead of 7 is used. However, the exponent bias is still a fixed value in the entire study. That is, the exponent bias is never dynamically adjusted to best fit the given input distributions of different layers of various models. In addition, only two formats (1,4,3) and (1,5,2) are adopted throughout the study. There are no discussions about exploring different combinations of the field size of exponent and fraction. Furthermore, the paper explicitly mentions that a notable accuracy loss is observed unless FP16 is used for the first layer and the last layer while performing inference of ResNet-18.
> Nevertheless, as explicitly mentioned in our paper, the proposed FFP8 format possesses three key features:
>
> A. The exponent bias can be ANY integer.
>
> B. The field size of exponent/fraction can also be adjusted.
>
> C. The sign bit can be presented or removed.
>
> These three features together achieve a remarkably low model accuracy loss (0.1\%~0.3\%) for tested models even the external weights/activations are only in 8-bit floating-point precision. In our opinion, the ideas proposed and the results achieved in this paper are far more than those introduced in (Sun et al., 2019).
>
> Last but not least, we have proposed a systematic transformation procedure to convert weights of a pre-trained FP32 model to a best-fit FFP8 model with extremely low accuracy loss. Yes, it is simple and needs no complicated TRAINING methodology. In our opinion, doing simple, doing less, but getting better results should be considered as an advantage, not a shortcoming, shouldn’t it?
>
> 2)
> In (Cambier et al., 2020), only FP8(1,5,2) format with a fixed bias b=15 is used. The authors proposed a “shift” operation and a “squeeze” operation to redistribute the weights and activations so that the value range after redistribution can be fit into the representable range window of FP8(1,5,2). The exact equation to do so is expressed as Equation-(1) on Page 4 of their paper. Note that every “shift operation” actually requires a floating-point multiplication. Moreover, a “squeeze” operation needs to transform an input value X to the output value Y = X^α, where Y, X, α are all floating-point numbers. There are obviously no low-cost means, neither hardware-based nor software-based, to carry out such “squeeze” operation, i.e., “squeeze” is an expensive (either hardware-hungry or runtime-consuming) operation.
> Alternatively, to support our proposed FFP8 format, only a simple low-cost FFP8 to FP32 hardware converter, illustrated in new Figure 6(c) of the revised manuscript, is required. As well, there is virtually no performance loss in a pipelined hardware design. In short, it is much easier to support FFP8 in a hardware perspective as compared to S2FP8 in (Cambier et al., 2020).
>
> << Summary >>
> Advantages over (Sun et al., 2019):
>
> A. The proposed FFP8 format is much more flexible: the presence of the sign bit, flexible field size for exponent and fraction, and ANY integer can be set as the exponent bias. It is the key to achieve a higher model accuracy. None of them was mentioned in (Sun et al., 2019).
>
> B. More accurate inference results.
>
> Advantages over (Cambier et al., 2020):
>
> A. Only FP8(1,5,2) is used in (Cambier et al., 2020), no flexibility on the format selection.
>
> B. No low-cost HW/SW solutions to the implementation of the “squeeze” operation in S2FP8, whereas FFP8 can be easily supported in HW with minimal extra logic.
>
> Once again, there is no doubt that (Sun et al., 2019) and (Cambier et al., 2020) are two excellent papers. However, our study still adds several new technical advances that have never been presented and discussed, which consequently leads to a more accurate inference outcome. Moreover, we have also presented a low-cost hardware solution to carry out all of the proposed ideas to show that our ideas are practical and implementable.
>
> ====================
>
> At last, we sincerely hope that the reviewer could kindly take a second look and re-evaluate the value of this paper. Thank you very much.

---

> ### Author Response · Authors · 2020-11-17
> **Reply to AnonReviewer 1 Part 2**
>
> Q2:
> On the same note, this paper claimed advantages of using FP32 computation engine, however, the three training papers, by definition, can also be used in FP32 computations through the same high precision converting described in this paper. In addition, it is not clear to me how much one can benefit from quantization while keeping computation in 32bit? For example, one may save some memories space, but the whole inference will be limited by the FP32 computations in terms of throughput, latency, power and cost. Can the authors provide an application case where only quantizing data is beneficial?
>
>
> A:
> We would like to thank the reviewer for bringing up these issues. We elaborate more on the following two issues: “how much one can benefit from quantization while keeping computation in 32bit” and “the whole inference will be limited by the FP32 computations in terms of throughput, latency, power and cost”.
>
> 1)
> Using a shorter format (8-bit or 16-bit) for loading/storing data from/to the external memory while still keeping the internal computation in 32-bit precision is exactly the current technology trend. The primary reason is that the actual performance bottleneck in today’s GPU servers is the limited external memory bandwidth, not the computation capacity. Therefore, using shorter data for external memory accesses greatly helps relieve the performance bottleneck while using FP32 in the internal computation cores helps maintain high computation accuracy.
>
> Here we list three documents for the reviewer’s reference.
> Document 1: Nvidia Tesla V100 GPU Architecture, officially released by Nvidia,
> [ https://images.nvidia.com/content/volta-architecture/pdf/volta-architecture-whitepaper.pdf ]
>
> Figure 8 on Page 15: V100 supports “Tensor Core 4x4 Matrix Multiply and Accumulate”, i.e., D = A x B + C, where A~D are all 4x4 matrices; C and D are in FP32, A and B are in FP16. This kind of computation core is also referred as fused-multiply-accumulate (FMA) unit.
>
> Figure 9 on Page 16 explicitly points out that input data from external memory are in FP16 and the internal MAC computations are in FP32.
>
> Document 2: Nvidia A100 Tensor Core GPU Architecture, officially released by Nvidia,
> [ https://www.nvidia.com/content/dam/en-zz/Solutions/Data-Center/nvidia-ampere-architecture-whitepaper.pdf ]
> Table 3 on Page 27 explicitly points out that up to 16x performance gain can be achieved if FP16 instead of FP32 is in use for input data on A100.
>
> Document 3: [ https://techdecoded.intel.io/resources/leadership-performance-with-2nd-generation-intel-xeon-scalable-processors/#gs.l4lz4j ]  from Intel
>
> Not only Nvidia, Intel also adopts the similar approach to boost the AI inference performance of the 2nd generation Xeon scalable processor with AVX-512 VNNI (Vector Neural Net Instructions) instruction set: one instruction can do 64 MAC operations, where inputs are in INT8 and the accumulation result is in INT32.
>
> The above three documents are from two industry giants: Nvidia and Intel. Hence, it should be conclusive that the combination of shorter external data + FP32 internal computation is both efficient and accurate. In this paper, what we would like to achieve is to further shrink the memory data from FP32 to FFP8 while maintaining the model accuracy.
>
> 2)
> Per the issue of “the whole inference will be limited by the FP32 computations in terms of throughput, latency, power and cost”, we have already clarified in the above item: both Nvidia and Intel point out that the performance bottleneck is the limited memory bandwidth instead of FP32 computation. Hence, an even shorter memory data format (e.g., FP32/FP16 to FFP8) can further boost the system performance and throughput significantly. Per the power issue, the external memory access is one of the major power consumption sources. Hence, the proposed 8-bit FFP8 format can also help reduce the energy consumption per a data access. Per the cost issue, the required external memory capacity can be reduced to just a quarter.

---

> ### Author Response · Authors · 2020-11-17
> **Reply to AnonReviewer 1 Part 1**
>
> Q1:
> It is not clear to me the objective of this paper. This paper introduces an 8-bit quantized inference framework. However, it is well-known that, 8-bit precision can be applied to popular DNN models to accelerate inference while maintaining model accuracies and many such systems have already been put into products (e.g. TPU). The state-of-the-art inference quantization work are primarily in the precision of 4 bit or less. But this paper did not compare their work to any of the latest inference works. Instead, the three references this paper compared to, i.e. (Wang & Choi, 2018; Cambier et al., 2020; Sun et al., 2019), are all for the acceleration of DNN TRAINING, where the challenging can be very different, e.g. quantization of gradient tensors.
>
> A:
> Thank you for the comment. Let us elaborate more on the reviewer’s concerns as follows.
>
> 1)
> The objective of this paper is clear: developing a better 8-bit floating-point format that achieves both accurate and memory-efficient inference.
> The reviewer mentioned about Google’s TPU as an example of 8-bit inference. However, please note that TPU only supports 8-bit INTERGER computations but not 8-bit floating-point computations. In most cases, the combination of 8-bit floating-point inputs plus 32-bit floating-point fused-multiply-add (FMA) units achieves higher accuracy than its integer counterpart.
> Meanwhile, the meaning of “while maintaining model accuracies” could be ambiguous since some papers consider the model accuracy is “maintained” even if an accuracy loss of 1\% to 2\% is actually presented. Nevertheless, in our paper, the accuracy loss is around 0.1% to 0.3% for VGG-16 and ResNet18/34/50. In our opinion, floating-point operations inherently outperform integer operations in terms of model accuracy.
>
> 2)
> As the reviewer mentioned, there are many quantization techniques are in 4-bit precision or even less (e.g., binarized models). Once again, these techniques still have to pay the price: a notable lower model accuracy. Recently, a NIPS-2019 paper, [ Ron Banner, Yury Nahshan, Daniel Soudry, Post training 4-bit quantization of convolutional networks for rapid-deployment ]
>  [ https://papers.nips.cc/paper/2019/file/c0a62e133894cdce435bcb4a5df1db2d-Paper.pdf ], has already investigated this issue in detail.
>
> 3)
> As the reviewer also mentioned, three previous studies (Wang & Choi, 2018; Cambier et al., 2020; Sun et al., 2019) are about 8-bit TRAINING. We are fully aware that these three studies are about training. As well, there is no doubt that training in 8-bit floating-point numbers is very challenging and the above three papers are all excellent in that specific topic.
> Indeed, this paper has nothing to do with 8-bit training. However, the OBJECTIVE of this paper is to achieve a more accurate INFERENCE result, and the proposed FFP8-based technique DOES provide more accurate inference outcomes than those three studies.
>
> << Highlights of the Response >>
>
> A. The objective of this paper is clear: developing a better and highly flexible 8-bit floating-point format that achieves both accurate and memory-efficient inference.
>
> B. It is inherent that FP computations achieves higher model accuracy than integer computations. Hence, models in 8-bit floating-point are required in additional to those models in INT8 if a higher model accuracy is desired.
>
> C. A model in 4-bit precision or less suffers a notable accuracy loss, which is a price cannot be avoided. Hence, it is absolutely not the target of this paper.
>
> D. (Wang & Choi, 2018; Cambier et al., 2020; Sun et al., 2019) are three excellent papers about DNN training in 8-bit FP numbers; however, the model using the proposed FFP8 format achieves more accurate inference outcomes than the models trained by those three studies.

---

### Official Review · AnonReviewer4 · 2020-10-29
**New flexible floating point format - review**

**Rating:** 7
**Confidence:** 3

**Review:**

The paper proposes a new flexible floating point format (FFP8) on 8 bits, to help alleviate the high memory demand of deep networks inference, while preserving high accuracy. There is a large body of literature on reducing the data format, typically from 32 bits to 16, 8 and even below. There is previous work on using an 8-bit floating point FP(8), usually (1,4,3) or (1,5,2) where 1 bit is used for sign, 5 or 4 bits are used for the exponent and 3 or 2 bits are used for the fraction.

The new FFP8 proposed in the paper offers more configurable parameters: (1) the bit width of exponent/fraction, (2) the exponent bias (usually this is implicit in FP8 but it can be changed in FFP8), (3) the presence of the sign bit (this can be removed and the bit can be used for exponent or fraction). By allowing flexible control of the dynamic range of the 8-bit FFP8, the accuracy loss is minimized. The authors observe that both the maximum magnitude and the value distribution are quite dissimilar between weights and activations in most DNNs. Therefore, the variable exponent size and exponent bias are better suited. Some activation functions always produce non-negative results (e.g., ReLU), therefore the sign bit can be reclaimed.

The paper includes experimental evaluation where a best fit format selection shows a minimal accuracy loss for the VGG-16 network. Further optimization can be achieved over each layer, because distributions across different layers are dissimilar for both weights and activations, and the distribution of an individual layer is typically narrower than the one of the entire model.

I advocate for the acceptance of the paper. Although there is a long line of research examining the data format for training and inference in deep networks, the paper provides evidence of the usefulness of some flexibility in the 8-bit format. The proposed would not require hardware changes except for simple translations between FP32 and FFP8.

Minor comments / typos:

Fig. 4 and 5 - the vertical axis might look better with the same max values (range).

pg.2 : Moreover, recent studies proposed several training frameworks that __producing__ weights only in 8-bit floating-point formats

pg.3 : Note that there is actually only one parameter, the bit width of the exponent (y), __that__ can be freely chosen when defining

---

> ### Author Response · Authors · 2020-11-15
> **Reply to AnonReviewer 4**
>
> Q1: Although there is a long line of research examining the data format for training and inference in deep networks, the paper provides evidence of the usefulness of some flexibility in the 8-bit format. The proposed would not require hardware changes except for simple translations between FP32 and FFP8.
>
>
> A:
> Thank you very much for your encouraging comment. As you have precisely pointed out: the proposed FFP8 format can bring a set of advantages for inference in deep networks, and the only extra hardware cost is merely a simple converter between FP32 and FFP8. For your information, we will include a more detailed hardware block diagram in the revised manuscript to show how an FFP8 number is converted to its FP32 counterpart, as illustrated in [ https://imgur.com/a/CQam38l ]. Just like what you have mentioned, the hardware for the translation is simple.
>
> ====================
>
>
> Minor comments / typos:
>
>
> A: Thank you very much for pointing out these issues, which greatly helps us improve our manuscript further.
> 1) The max values of the vertical axes in Figure 4 and in Figure 5 are now identical.
> 2) Two grammatical errors on Page 2 and Page 3 have been corrected according to your suggestions.

---

### Official Review · AnonReviewer3 · 2020-11-02
**Promising idea but no end-to-end experiment**

**Rating:** 6
**Confidence:** 4

**Review:**

This paper presents a flexible floating-point format that can save storage space by being highly configurable in terms of the bit width of the exponent/fraction field, the exponent bias, and the presence of the sign bit. The experiments in the paper demonstrate that the proposed format achieves a very low accuracy loss of < 0.3% compared to the regular float32 format for several popular image classification models.

Strengths
---
- The concept of flexible floating-point format makes sense and can potentially result in significant space savings for large models.
- The loss drop in the models compared to the standard float32 format seems to be minimal
- The authors evaluated the proposed format over multiple models: VGG 16, ResNet-50, ResNet-34, and ResNet-18

Weaknesses
---
- No full system implementation to demonstrate the performance/memory gains. I that the updates to the form registers will be infrequent and cheap
- The hardware details are very sketchy. Figure 6 seems to be only showing the high-level components.


Overall, the paper’s ideas are promising but my score reflects the fact that the authors presented end-to-end experiments demonstrating the performance/storage gains.

---

> ### Author Response · Authors · 2020-11-15
> **Reply to AnonReviewer 3**
>
> Q1:
> No full system implementation to demonstrate the performance/memory gains. I doubt that the updates to the form register will be infrequent and cheap.
>
> A:
> Thank you for the comment about the performance/memory gains as well as the cost and the updating frequency of the “format register”. We address your concerns as follows.
>
> 1)
> When deep learning applications run on current CPU/GPU-based servers, the real performance bottleneck in most cases is actually the limited memory bandwidth, not the lack of computing resources. That is why Nvidia has developed NVLink, NVSwitch, and GPU Direct technologies to boost the external memory bandwidth and shorten the external memory latency. Therefore, storing activations and weights in a shorter format (32-bit to 8-bit) obviously reduces the memory bandwidth requirement significantly, which helps relieve the memory bottleneck and thus improves the overall system performance. One more evidence, recent Nvidia GPU cores start to support two new floating-point formats, 16-bit BFloat16(BF16 or BFP16) and 19-bit TensorFloat32(TF32). Again, it is a very clear sign that a shorter data format does help achieve better system performance. An official article released by Nvidia,
> [ https://blogs.nvidia.com/blog/2020/05/14/tensorfloat-32-precision-format/ ], which explicitly indicates that a shorter data format (FP32 to FP16) can accelerate the BERT training by 3x on V100.
>
> In summary, it is conclusive that an 8-bit floating-point format can undoubtedly deliver a significant performance and memory gains as compared to its 16-bit and 32-bit counterparts in today’s state-of-the-art GPU computing platforms.
>
> 2)
> The “format registers” are updated only when a new FFP8 format is selected for the incoming activations or weights. That is, the format registers are set ONLY ONCE if a deep learning application only adopts one format for activation and one format for weight throughout the entire application time. If the proposed layer-wise optimization (LWO) technique is in use, those “format registers” are updated ONLY at the beginning of each layer if necessary. Take VGG-16 as an example, there are more than 1.8 billion MAC operations in Layer 2. It suggests once the format registers are properly set, those settings remain unchanged and valid for the succeeding computational operations for a long while. Moreover, there are at least 90 million MACs in any layer of VGG-16. In summary, the update of format registers is indeed infrequent and the runtime overhead for setting the format registers is negligible.
>
> Besides, the update of format registers is achieved through setting new values to hardware register bits, which is considered not an expensive operation.
>
> Q2:
> The hardware details are very sketchy. Figure 6 seems to be only showing the high-level components.
>
> A:
> Thank you for your valuable comment.
> Due to the page limitation, we only put a very simplified block diagram in Figure 6(c) of our initial manuscript. According to your suggestion, more hardware details of the proposed converter are revealed as shown in [ https://imgur.com/a/CQam38l ], and the manuscript will be further revised to include this new figure accordingly.
>
> As illustrated above, the extra hardware merely consists of few register bits, few logic gates, a mux-based data-bit selector, a subtractor, and an adder. An example of conversion from FFP8 to FP32 is also included. It should be clear enough that the size of the proposed converter is indeed much smaller than that of a 32-bit floating-point fused-multiply-add (FMA) unit.
>
> Let us emphasize once again that the update of the format register is infrequent. Once set, all succeeding activation and weight values from the external memory are simply 8-bit FFP8 data transfers, as the figure demonstrates.

---

### Decision · Program_Chairs · 2021-01-07
**Final Decision**

**Decision:**

Reject

**Comment:**

After reading the paper, reviews and authors’ feedback. The meta-reviewer agrees with the reviewers that the paper has limited novelty as there are already previous studies on setting floating point configurations. Additionally, the particular hardware setting that the authors provide seems to rely on a fp32 FMA, which defeats the purpose of a low bit floating point(where smaller FMA could have been used).Therefore this paper is rejected.

Thank you for submitting the paper to ICLR.